# Biophysical Characterization of (Silica-coated) Cobalt Ferrite Nanoparticles for Hyperthermia Treatment

**DOI:** 10.3390/nano9121713

**Published:** 2019-12-01

**Authors:** Niklas Lucht, Ralf P. Friedrich, Sebastian Draack, Christoph Alexiou, Thilo Viereck, Frank Ludwig, Birgit Hankiewicz

**Affiliations:** 1Institute of Physical Chemistry, Universität Hamburg, Grindelallee 117, 20146 Hamburg, Germany; 2Department of Otorhinolaryngology, Head and Neck Surgery, Section of Experimental Oncology and Nanomedicine (SEON), Else Kröner-Fresenius-Stiftung-Professorship, Universitätsklinikum Erlangen, Glückstraße 10a, 91054 Erlangen, Germany; 3Institute for Electrical Measurement Science and Fundamental Electrical Engineering, Technical University of Braunschweig, Hans-Sommer-Straße 66, 38106 Braunschweig, Germany

**Keywords:** magnetism, hyperthermia, Induction, magnetic fluids, toxicity

## Abstract

Magnetic hyperthermia is a technique that describes the heating of material through an external magnetic field. Classic hyperthermia is a medical condition where the human body overheats, being usually triggered by a heat stroke, which can lead to severe damage to organs and tissue due to the denaturation of cells. In modern medicine, hyperthermia can be deliberately induced to specified parts of the body to destroy malignant cells. Magnetic hyperthermia describes the way that this overheating is induced and it has the inherent advantage of being a minimal invasive method when compared to traditional surgery methods. This work presents a particle system that offers huge potential for hyperthermia treatments, given its good loss value, i.e., the particles dissipate a lot of heat to their surroundings when treated with an ac magnetic field. The measurements were performed in a low-cost custom hyperthermia setup. Additional toxicity assessments on Jurkat cells show a very low short-term toxicity on the particles and a moderate low toxicity after two days due to the prevalent health concerns towards nanoparticles in organisms.

## 1. Introduction

Magnetic fluids have proven to offer a wide variety of applications in material sciences [1], physics [2], and medicine [3,4]. Especially, the latter still have substantial concerns regarding the toxicity of ferrofluids, i.e., nanoparticle suspensions, in general [5]. In a typical ferrofluid, iron oxides, as well as different transition metals, such as cobalt are used [6]. Cobalt ferrite offers a variety of advantages when compared to other typical magnetic phases, e.g., magnetite, in synthesis and in their respective magnetic behavior [7]. The synthesis of magnetite bears the inherent disadvantage that commonly byproducts, like maghemite, are formed during synthesis; oxidation to hematite is also a common problem. The magnetic properties of cobalt ferrite and magnetite are very similar in regard to their respective saturation magnetization. Due to the high magnetocrystalline anisotropy of the hard-magnetic cobalt ferrite, the Néel relaxation is suppressed at small particle sizes when compared to magnetite [8].

This work presents an insight into the toxicity of an aqueous cobalt ferrite suspension in comparison to its potential use in therapeutically applied magnetic hyperthermia. Each drug must accumulate in sufficient concentration to the target in order to achieve an effective destruction of malignant tissue [9]. However, this work does not focus on drug delivery [10], but on the magnetic behavior of the nanoparticles. Hence, it must be kept in mind that, given a working targeting mechanism, the particle concentration at the site and in the vicinity of the malignant tissue will always be far higher than in the blood or in healthy tissue.

## 2. Materials and Methods

The cobalt ferrite nanoparticles were synthesized through a modified synthesis that was derived from Nappini et al. [11]. 7.6 g cobalt(II)-nitrate (Sigma-Aldrich, St. Louis, MO, USA) and 17.3 g iron(III)-chloride (Sigma-Aldrich, St. Louis, MO, USA) were dissolved in 100 mL Millipore water. 2 mL concentrated nitric acid was added and the solution was heated to the boiling point. Incidentally, 400 mL of 1 M NaOH (Grüssing GmbH, Filsum, Germany) solution were prepared and then heated to the boiling point. The boiling salt solution was then rapidly transferred into the NaOH solution under vigorous stirring. The temperature was held for another 90 min. before cooling down the solution to room temperature. The particles were separated through magnetic decantation and then washed twice with water and suspended in 40 mL 2 M HNO_3_ (Merck KGaA, Darmstadt, Germany). The suspension was then heated to the boiling point and a boiling mixture of 56 mL (0.5 M) iron(III)-chloride and 28 mL (0.5 M) cobalt(II)-nitrate was rapidly added under vigorous stirring. After 30 min., the solution was washed again with water through magnetic decantation until a neutral pH was reached. The particles were then dispersed in 25 mL 0.25 M aqueous tetramethylammonium hydroxide (VWR LLC., Radnor, PA, USA) solution and slightly stirred overnight. The mass percentage of the particles was determined through gravimetry. For the final stabilization step, the particle solution was added to a 100 mM solution of citric acid (Grüssing GmbH, Filsum, Germany) with a target weight percentage of about 0.25 wt% and then stirred for at least 2 h. After another magnetic separation step, the particles were dispersed in the same volume of 20 mM trisodium citrate (Fluka Inc., Buchs, Switzerland) solution and then left on a stirrer for another hour. Lastly, the volume was reduced by about 75% through evaporation and the concentrated solution dialyzed against water for one week. The weight percentage of the final product was once again determined through gravimetry.

The synthetic pathway that was described by Zou et al. [12] was used for the silica coating. An equivalent of 50 mg citrate-stabilized particles was used and diluted to 100 mL with Millipore water, 0.5 mL hydrazine (Sigma-Aldrich, St. Louis, MO, USA), and 0.4 mL tetraethyl orthosilicate (Fluka Inc., Buchs, Switzerland) were added and heated to 90 °C for 2 h under reflux and stirring. The concentration of the final product was then adjusted by the evaporation of the solvent.

The pNipam beads were separately synthesized to the nanoparticles. Particles and gel beads were mixed overnight to produce the composite material. The gel beads were synthesized in a one-pot synthesis by dissolving 1 g *N*-isopropylacrylamide (Tokyo Chemical Industry Co. Ltd., Tokyo, Japan), 10.6 µL glutaraldehyde (Sigma-Aldrich, St. Louis, MO, USA), and 3.8 mg sodium dodecyl sulfate (SDS) (Carl Roth GmbH + Co. KG, Karlsruhe, Germany) in 40 mL Millipore water [13]. The solution was purged with nitrogen for 60 min. and heated to 80 °C in the meantime. Adding 21 mg potassium persulfate (Merck KGaA, Darmstadt, Germany) that was dissolved in 10 mL Millipore water started polymerization. The particle suspension was then dialyzed for seven days against water to remove the SDS and the volume was reduced by a rotary evaporator [14].

The static magnetic characterization was directly conducted in solution while using the commercially available EZ9 VSM (Microsense LLC, Tempe, AZ, USA) in a range of ±2.5 T.

The dynamic magnetic characterization was performed combining the spectra that were measured with two custom-built AC susceptometers (ACS) (low-frequency setup covering a frequency range from 10 HZ–10 kHz and high-frequency system that covered a frequency range from 200 Hz–1 MHz) [15,16].

Toxicity assays were carried out while using Jurkat cells, a human T-lymphocyte cell line (ATCC). 3.0 × 10^4^ Jurkat cells were seeded into 96 well plates and then treated with different amounts of nanoparticles or the same volume water as a negative control. The positive controls were treated with 2% DMSO (Carl Roth GmbH + Co. KG, Karlsruhe, Germany). After incubation for 24 and 48 h, 50 μL aliquots were treated with 250 μL freshly prepared staining solution consisting of 1 μL AnnexinA5-FITC conjugate (AxV) (Thermo Fisher Scientific, Waltham, MA, USA), 10 μg Hoechst 33342 (Hoe) (Thermo Fisher Scientific, Waltham, MA, USA), 66.6 ng propidium iodide (PI) (Sigma-Aldrich, St. Louis, MO, USA), and 0.4 μL/mL DiIC1(5) (1,1′-dimethyl-3,3,3′,3′-tetramethylindodicarbocyanine iodide, DiI, 10 μM) (Thermo Fisher Scientific, Waltham, MA, USA) per 1 mL Ringer’s solution (Fresenius Kabi AG; Bad Homburg, Germany) for 30 min. at 4 °C [17]. Apoptotic and necrotic cells were detected with AxV and PI, as well as Hoe, were used to identify the cells by nucleus staining and DiI staining was added to determine the integrity of the mitochondrial membrane potential. Flow cytometry was performed with a Beckman (Beckman Coulter Inc., Brea, CA, USA) Gallios™ cytofluorometer. Electronic compensation was used in order to avoid fluorescence breakthroughs. The data were analyzed with the help of the software Kaluza™ V. 1.2 from Beckman Coulter.

Hyperthermia was measured with a custom-built coil setup. The frequency was set while using the STEMlab web interface of the redpitaya^®^ (Red pitaya d.d, Solkan, Slovenia) amplified by a t.amp^®^ TSA 2200 (Thomann GmbH, Burgebrach, Germany) audio amplifier powering a series resonant circuit, including an additional 2 Ω power resistor. The factual field strength was directly calculated from the measured current driving the excitation coil. The current was measured with a Sensitec 3025 sensor (Sensitec GmbH, Lahnau, Germany). The signal was processed within STEMlab of the redpitaya^®^. The temperature was measured with a fiberoptic gallium arsenide measurement device from Optocon (Weidmann Technologies Deutschland GmbH, Dresden, Germany). The sample holder is a three-dimensional (3D)-printed ABS piece to fit 5 mL rotilab^®^ glass vials from Carl Roth (Carl Roth GmbH + Co. KG, Karlsruhe, Germany) and it is thermally insulated by a 2 mm water-cooled polystyrene hose.

The dynamic light scattering (DLS) measurements were performed with an ALV^®^/CGS-3 Compact Goniometer-System (ALV-Laser Vertriebsgesellschaft m-b.H., Langen, Germany) while using an ALV^®^/LSE-5004 Multiple Tau Digital Correlator (V.1.7.9.) in combination with a COBOLT™ SAMBA™ 50 laser (Cobolt AB, Solna, Sweden) (Nd:YAG, 532 nm, 400 mW) and the ALV^®^ Digital Correlator Software 3.0. The measuring angle was set to 90° for all measurements and every individual measurement was conducted for 60 s. The sample vials consisted of quartz glass and they were placed into a measurement cell filled with toluene. The temperature-dependent viscosity and refractive index of the solvents were automatically corrected, according to tabulated values. The toluene bath and, thus, the samples were tempered by a JULABO^®^ F25 (JULABO GmbH, Seelbach, Germany) thermostat working with a mixture of water and ethylene glycol and delivering a temperature accuracy of 0.01 °C.

## 3. Results

The particles presented in this work that are shown in Figure 1 have a core diameter of roughly 18 ± 4 nm with a coarse surface, which is formed during synthesis to increase the active surface of the particles and, thus, coordinate more citrate ligands to increase the stability. The size of the silica shell can be adjusted by the precursor concentration in the coating step. In this work, only silica coated core-shell particles with a shell thickness of 5 nm were used as a proof of principle, as the focus was on comparing the pure cobalt ferrite with ‘sealed’ cobalt ferrite to assess whether the particles themselves have toxic effects or the surface properties are more responsible for the appearing toxicity.

### 3.1. Magnetism

Figure 2 shows the static magnetization curves of both the citrate stabilized and silica coated particles. The cobalt ferrite has a high saturation magnetization of about 52 Am^2^/kg, which is comparable to other particle systems in our size regime [7,18,19,20,21]. The saturation magnetization of the silica-coated cobalt ferrite particles is significantly lower than that of the CoFe_2_O_4_ particles if the measurements are normalized to the weight fraction of the dissolved particles. The normalization of both curves to their respective ferrite content results in the same saturation magnetization, as expected.

The imaginary part of the AC susceptibility (ACS) within the Debye model is expressed by:(1)$$\chi^{\prime \prime }{=\;\chi }_{0}\frac{\omega \tau }{1\;+\;(\omega \tau )\text{²}}$$

The position of the peak maximum of the imaginary part from the ACS-spectra can be obtained by the derivative of Equation (1) that is solved for the frequency, leading to Equation (2).
(2)$$f_{\max}=\;\frac{1}{2\pi \tau }$$

As described by Draack et al. [22], the particle system in this work is dominated by the Brownian relaxation mechanism, which is important for an effective heating in hyperthermia [8,23]. From Equation (1), the imaginary part of the ACS is at maximum at *ωτ* = 1, i.e., the Brownian relaxation time can be calculated from the frequency of the *Χ″* maximum. The hydrodynamic radius of the particles can be calculated utilizing Equation (3) while assuming purely spherical particles.
(3)$$r_{H}=\sqrt[{3}]{\frac{\tau_{B}k_{B}T}{4\pi \eta }}$$

The data are in good agreement with the measurements of the geometric radius of the particles that were derived from transmission electron microscopy (TEM). The particles appear to be slightly bigger with 21 nm in diameter (as compared to 18 nm from TEM) in the ACS (Figure 3), suggesting a solvate layer of about 1.5 nm. The maximum of the imaginary part from the ACS also shows the ideal excitation frequency for hyperthermia, which in this case is at around 19 kHz at zero field [16].

This frequency can be optimized for higher field strengths, as depicted in Figure 4. The plot visualizes the theoretical shift of the maximum of the imaginary part from the ACS data from Figure 3 with increasing field strength. At 25 mT a rough increase of approximately a factor four to about 72 kHz is to be expected. The plot was realized utilizing an approximation (Equation (4)) from Yoshida et al. [24] based on solving the Fokker–Planck equation, which was experimentally verified [25,26]:(4)$$\tau_{B}=\;\frac{\tau_{0}}{\sqrt{1\;+\;0.126\xi^{1.72}}}$$

This equation estimates the decrease of the relaxation time of a particle system with increasing field strength due to the nonlinear behavior of the Brownian relaxation. *ξ* is defined as the quotient of the magnetic energy and the thermal energy, as shown in Equation (5). Here, *m*_p_ denotes the magnetic moment of a single particle, which was derived from the saturation moment that was estimated from static magnetization measurements and the core volume of the particles from TEM and *B* is the magnetic flux density.
(5)$$\xi =\;\frac{m_{p}B}{k_{B}T}$$

The plot shown in Figure 4 is described with the consolidation of Equations (4) and (5) and it can be seen in Equation (6).
(6)$$\frac{f(B)}{f\left(B\;=\;0\right)}=\;\sqrt{1\;+\;0.126{\left(\frac{mB}{k_{B}T}\right)}^{1.72}}$$

### 3.2. Viability

Flow cytometry analyzed the cell viability after incubation of Jurkat cells with the two particle systems. Figure 5 shows the induced cytotoxicity of both cobalt ferrite systems. During the first 24 h, the presence of both systems was not affecting the cellular viability up to a concentration of 400 µg/mL (Figure 5A,B). After 48 h, the measurements revealed no negative effects up to 100 µg/mL. At higher concentrations, beginning with 200 µg/mL, the particles start to cause dosage dependent toxicity (Figure 5C,D).

Hence, both of the systems show a similar behavior being nontoxic below 200 µg/mL with a significant toxicity above. Interestingly, both of the systems express a slight difference of toxicity, which suggests that the differences in toxicity seen after 48 h are due to the direct cellular exposure of cobalt ferrite. This exposure is apparently shielded by the silica shell of the CoFe_2_O_4_@SiO_2_ particles, leading to a reduction in particle-induced cell stress. Further tests must be performed to gain a better insight into the mechanism of toxicity and to further distinguish cytotoxic effects, cellular pathways, and ion release from pure particles and core-shell particles [27].

## 4. Discussion

### 4.1. Viability

Both particle species are not very stable in cell culture medium, which leads to particle sedimentation and, thus, to a very high particle concentration on and in the immediate vicinity of the cells at the bottom of the cell culture well. Nevertheless, our system shows a general low toxicological impact, which is astonishing given the sheer number of particles in the vicinity of the cells (Figure 6). Bossi et al. [28] showed that cobalt oxide (Co_3_O_4_) below 50 nm can migrate into the cell plasma and it is not dependent on the endocytosis pathway. While this is a convenient feature for drug delivery, it can lead to increased cellular toxicity, as shown on kidney cell DNA for cobalt ferrite particles with a rough size of 40 nm in concentrations below 100 µg/mL [29]. Pure cobalt nanoparticles of about 50 nm can also inhibit mRNA and protein expression and even alter cell morphology [30]. Another example of the effect of cobalt ferrite particles in a size regime between 30 and 50 nm shows a significant decrease in blood cells and a strong enrichment in vital organs [9]. Cobalt ferrite particles that are below 10 nm can easily intracellularly and perinuclearly accumulate while expressing little or no inhibition of cell proliferation or general toxic effects in low concentrations [31]. It should also be noted that there is a general risk that Co^2+^-ions could induce hypoxia in the human body if the particles degrade during their exposure time [32]. Wang et al. [33] presented a Co_3_O_4_ particle system in a similar size regime to ours between 15 and 30 nm and found an already severely inhibited cell growth at concentrations between 12.5 to 200 µg/mL, an impaired mitochondrial membrane potential, but no obvious cell death and accumulation of reactive oxygen species.

### 4.2. Hyperthermia:

When applying an alternating magnetic field, the particle movement generates friction in its respective solvent, which results in heating power. This heat can be assessed in a calorimetric temperature increase vs. time measurement. The generated heat is referred to as the loss power, or specific loss power (SLP) when normalized to the respective particle mass used in the measurement in watts per gram of particles. Equation (7) describes the SLP:(7)$$SLP\;=\;\frac{c}{m}\frac{\Delta T}{\Delta t}$$

Utilizing the heat capacity *c* of the solvent, here water, and the mass of magnetic material *m*. The temperature vs. time curve typically starts with a linear region, followed by a saturation effect. Normally, the linear slope is used to calculate the SLP. Although this equation does not consider the external field parameters, they must nonetheless be considered [34]. In general, as the field strength increases, the induced heat increases. Starting from low field strengths, the induced heat linearly increases until a saturation regime is reached based on the actual static magnetization of the system. In terms of practical construction, it should be noted that, with increasing field strength achieved by increasing the current amplitude of the excitation coil, the passive heat that is generated by the resistance of the coil wire also increases (via *P* = *I*^2^ Re{*Z*}). In an actual application, this problem can be circumvented by an effective cooling system. In our small laboratory setup, we measured between 12.2 mT and 24.4 mT field strengths, which yielded good results while also keeping the error from coil-heating as low as possible. Figure 7 shows the heating vs. time curves of three different field strengths. The error that is caused by the passive heating is depicted as colored clouds. At the lowest field strength, which corresponds to 2.5 A current, there is basically no coil heating over a long time period, even far outside the time that is shown in the graph.

Even the highest field strength that corresponds to 10 A current has little to no error caused by coil heating in the first 180 s in the respective linear regime. Additionally, the field frequency is decisive. If the frequency is too low, the particles generate low friction in the solvent and, thus, limit the effectiveness of the heating. If the frequency is too high, the particles can no longer follow the field. ACS (Figure 3)—corrected for the field-dependence of the Brownian relaxation time—can be utilized to assess the optimal frequency while using the peak value of the imaginary part. Figure 8 shows the SLP in dependence of the magnetic field strength. The values follow a nearly perfect linear trend in the measurement window. At high field strengths, this curve is expected to saturate. Even at very low fields of 12 mT, the particles can already absorb 20 W of power per gram nanoparticles, which is a very good value when compared to the literature [20,21,35,36].

### 4.3. Outlook

ACS measurements shall be performed at different field strengths and the hyperthermia device will be modified to fit the ideal excitation frequency for a more effective heat generation of the particles. The viscosity of the particle carrier medium also plays an important role for the excitation frequency. A higher viscosity at our given field frequency results in a higher excitation efficiency due to the viscosity induced slowdown of the Brownian relaxation time yielding a shift of the ACS peak frequency to lower the frequencies. A low frequency of 10 kHz could already induce a high temperature increase in a composite material, i.e., pNipam beads with CoFe_2_O_4_ in an aqueous environment. pNipam has the side effect of being thermoresponsive, i.e., it is well dispersed and swollen below the lower critical solution temperature and it undergoes a steep phase transition to hydrophobic, resulting in rapid shrinkage of the beads as the water is forced out of them.

Figure 9 shows the temperature dependent radii of the pNipam beads in water, which rapidly decreases when the lower critical solution temperature (LCST) is reached. This sample was then mixed with different amounts of our cobalt ferrite particles and measured in the hyperthermia setup (Figure 10). A content of about 2 wt% of the total mass enables a rapid heating of the material to induce the phase transition in about 100 s, which is illustrated by two different cobalt ferrite concentrations.

This exemplarily shows how the matrix is influenced by an ac-field that paves the way to another application area that is currently being further researched for potential future application in a material science direction. Polymer networks (micelles, beads) are often discussed as a carrier to release a drug at a specified target location. The main hurdle is the complex release mechanism. In our system, the release can be triggered by inducing the phase transition of the pNipam with a locally applied ac-field. Regarding the biocompatibility further steps are also required. The particles must be optimized in their hemocompatibility, to stabilize them in serum and, ultimately, blood to prevent agglomeration. In a next step, the opsonization behavior of the particles in serum and blood has to be examined, as opsonization has significant influence on a targeting mechanism of a drug.

## 5. Conclusions

In this work, we were able to present a proof-of-principle for a presumably superparamagnetic nanoparticle system with an acceptable toxicological footprint and high net magnetization. The particles are highly stable in aqueous solution but they tend to precipitate quite quickly in cell media and even faster in human blood. The particles have high heat generation in ac magnetic fields, which can be further optimized by frequency tuning of the generator. Our particle system is a convenient model system for multiple possible medical applications due to its high loss power in non-contact heating and in technical applications, due to its ability to induce phase transition in responsive materials.

## Figures and Tables

**Figure 1 nanomaterials-09-01713-f001:**
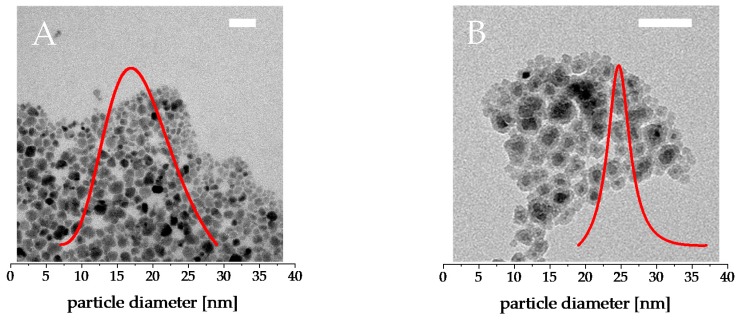
Transmission electron microscopy (TEM)-micrographs of citrate stabilized cobalt ferrite nanoparticles (**A**) and silica coated core-shell particles (**B**). The inlays depict the particle size distribution of the particle species with 18 ± 4 nm for the cobalt ferrite particles and 27 ± 4 nm for cobalt ferrite@SiO_2_. The size distribution fits well to an expected log normal distribution; the pictured distributions are normalized to one. The white scale bars depict 50 nm each.

**Figure 2 nanomaterials-09-01713-f002:**
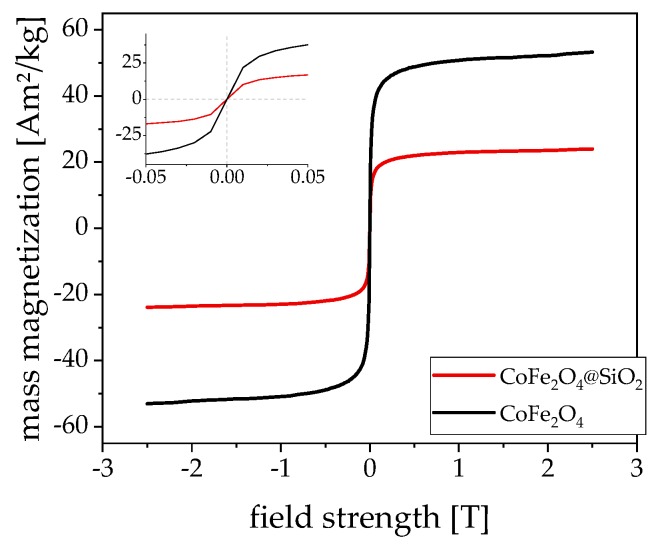
Static magnetization curve of CoFe_2_O_4_ (black) and CoFe_2_O_4_@SiO_2_ (red) normalized to the particle mass, not the magnetic content. The inlay zooms in on the zero passage, which shows no apparent remanence at room temperature.

**Figure 3 nanomaterials-09-01713-f003:**
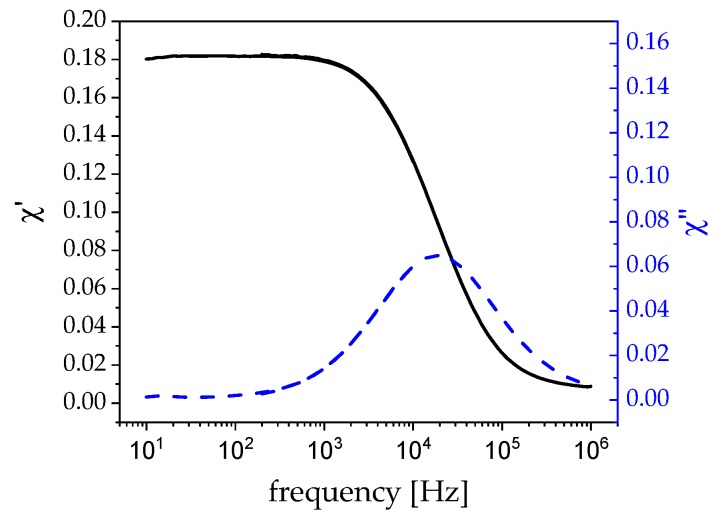
AC susceptibility measurement of 5 wt% CoFe_2_O_4_ in watery solution divided in real part (black) and imaginary part (blue).

**Figure 4 nanomaterials-09-01713-f004:**
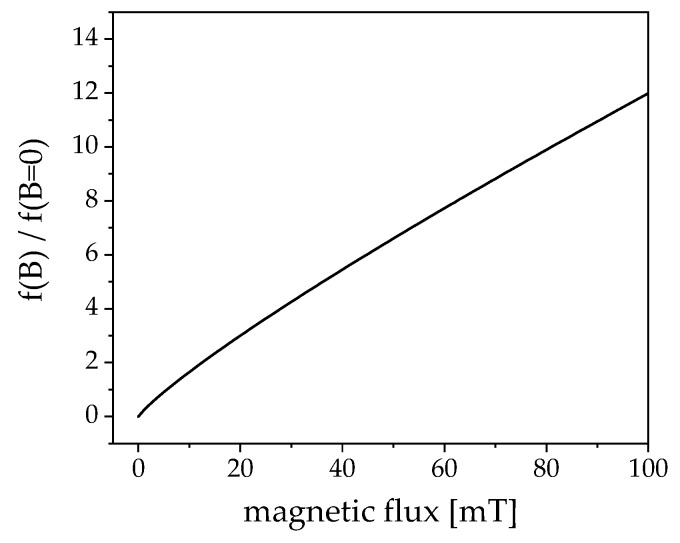
Approximated factorial shift of the peak value of the imaginary part in ACS with increasing ac magnetic field density. A magnetic flux of roughly 25 mT leads to a rough quadrupled frequency for the peak value.

**Figure 5 nanomaterials-09-01713-f005:**
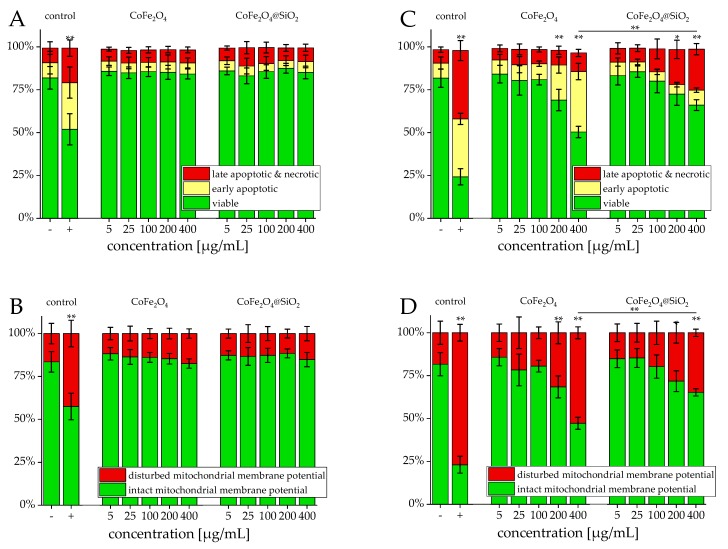
Viability of Jurkat cells after incubation with CoFe_2_O_4_ and CoFe_2_O_4_@SiO_2_ particles. Cells were incubated for 24 h (**A**,**B**) and 48 h (**C**,**D**) with increasing amounts of particles and analyzed by multiparameter flow cytometry. Viability was determined by AxV–FITC and PI staining (first row), yielding the percentage of viable (Ax− PI−), early apoptotic (Ax+ PI−), and late apoptotic and necrotic (PI+) cells. The status of the mitochondrial membrane potential was analyzed by DiIC_1_(5) staining and distinguishes cells with intact (DiIC_1_(5) positive) and depolarized (DiIC_1_(5) negative) membranes (second row). Positive controls contain 2% DMSO, and negative controls represent the corresponding amount of solvent instead of ferrofluid. The depicted concentrations are normalized to their respective particle concentration, not the magnetic content. Statistical significance of viability and intact membrane potential between negative control and samples and between the two particle systems are indicated with * *p* < 0.005 and ** *p* < 0.0005 and were calculated via Student’s t-test analysis with *n* = 8 for each shown sample.

**Figure 6 nanomaterials-09-01713-f006:**
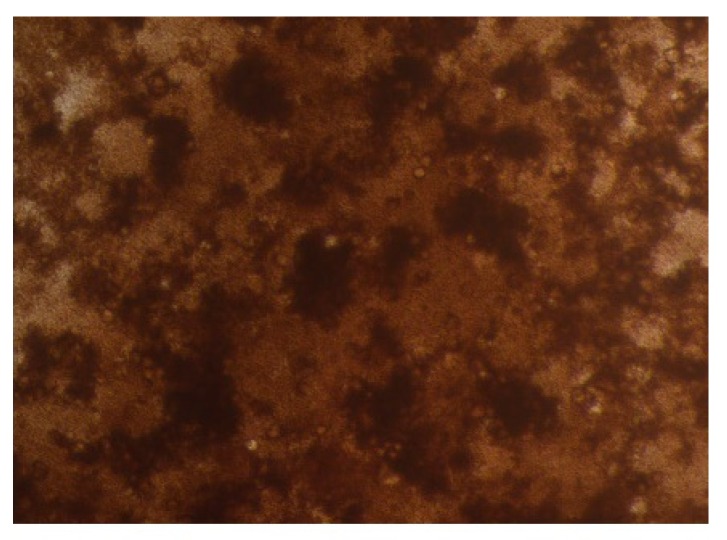
Photograph of a Jurkat cell culture incubated with 400 µg/mL CoFe_2_O_4._

**Figure 7 nanomaterials-09-01713-f007:**
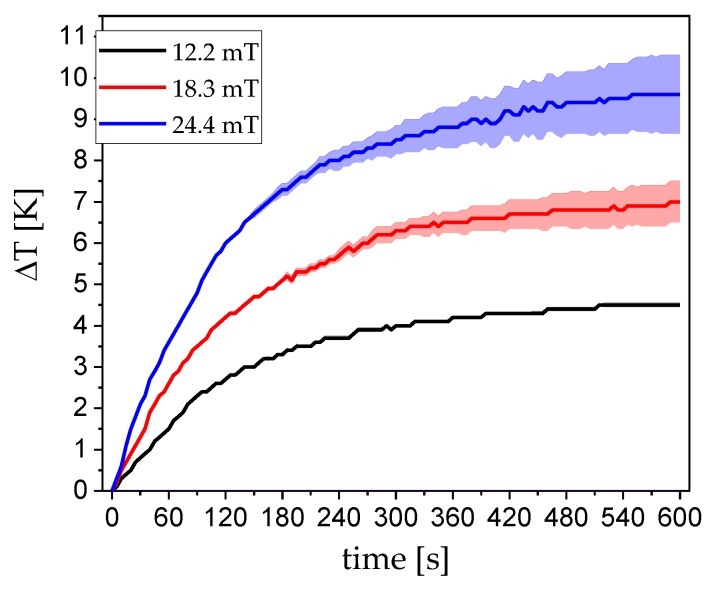
Field dependent heating vs. time curves of 1 mL 1 wt% CoFe_2_O_4_ at 10 kHz. The colored clouds around the graphs depicts the error produced by coil heating.

**Figure 8 nanomaterials-09-01713-f008:**
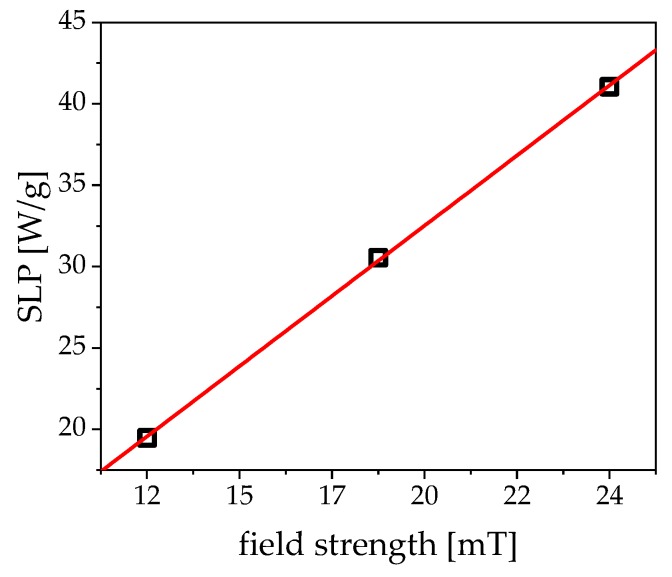
Calculation of the field dependent specific loss power (SLP) from the data of Figure 7. The red line depicts a fit function, estimating the linear dependence of the SLP to the field strength.

**Figure 9 nanomaterials-09-01713-f009:**
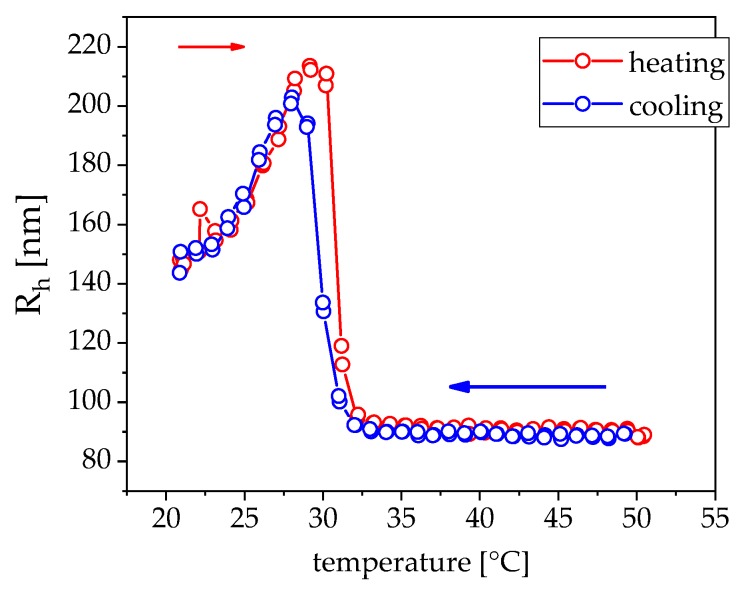
Hydrodynamic radius of pNipam gel beads in water mixed with 2.8 wt% CoFe_2_O_4_ (red line in Figure 10) in dynamic light scattering; the sample is first heated and then cooled down. The phase transition temperature of the gel is about 30 °C with a very small hysteresis.

**Figure 10 nanomaterials-09-01713-f010:**
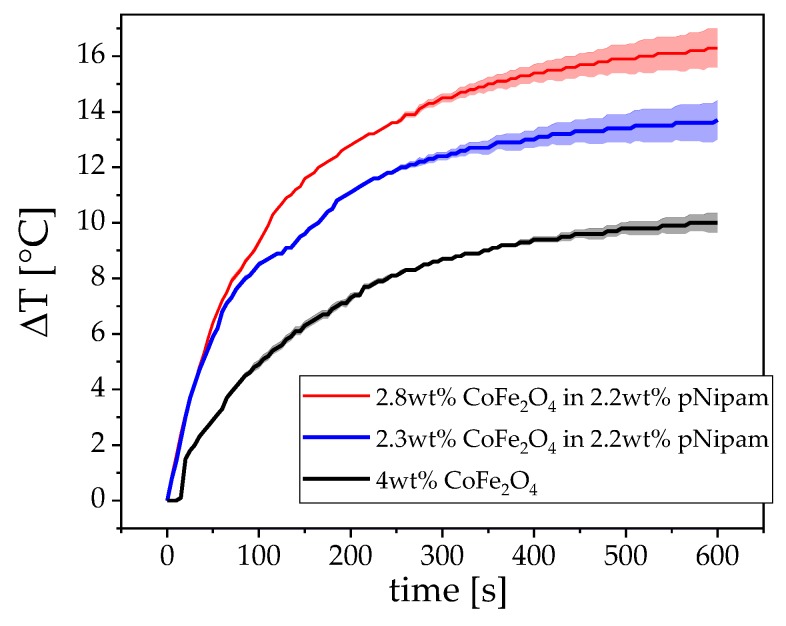
Temperature vs. time curves of different composites CoFe_2_O_4_ and pNipam gel beads in aqueous solution. The colored clouds depict the error from coil heating.

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
