# Peer review of "Biophysical Characterization of (Silica-coated) Cobalt Ferrite Nanoparticles for Hyperthermia Treatment"

_nanomaterials, 2019, doi:10.3390/nano9121713_

Round 1

Reviewer 1 Report

Dear Authors,

thank you for your interesting work on cobalt-ferrite nanoparticles for thermotherapy. I agree with the fact that these compounds can improve efficiency compared to pure iron oxide ones.

Nevertheless, some aspects should be precised:

What kind of system targeting are you looking for your particles ? Local deposition (eg by needle puncture) ? Systemic administration ? In this case, what are you envisaging to avoid opsonization and margination of your particles ?
- line 57: can you precise the molarity of the iron and cobal chloride solutions ?
- Please, be homogenous in your TM and R adds. All the tradenames shoud be correctly noted. (ie line 93 GalliosTM cytofluorometer and not Gallios cytofluorometerTM, line 97 RedPitayaTM instead of redpitaya)
- Line 207: Colloid unstability in cell cultur medium: is it only magnetic unstability (ie are your silicate - and citrate-coated particles stables for a long period of time in native suspension media ?) or are you thinking about opsonization effet ?
- Line 152: the discrepancy between TEM and magnetization curve derived particle sizes can be also explain by the influence of the polydispersity (even if in your case, it is rather low), and different ponderation factor on size-class averaging. Can you some words to complete your explanation ?
- Line 191: what is the number of the Jurkat cell experiment samples (n= ????) ? Have you shown the normality of the distribution ? If not, please use a non-parametric test (and not undefined "Student's t-test").
- Figure 5A: can you comment the fact that negative control is rather highly toxic at 24 h for induction of apoptosis ?
- I do not really understand the interest of the use of PNIPAM gel beads experiments in your logical demonstration. Of course, the figure 9 shows the ability of heating PNIPAM beads by your magnetic particles but are you meaning that PNIPAM suspensions would be adequate media for local particle administration ? Most of the time, it is the opposite that is envisaged (fluidity at ambiant temperature for injection, gelification at 37°c body temperature to increase local residency time) ? Please clarify in the text.
- line 288: not paramagnetic particles but (eventually, not really proved in this woek) superparamagnetic ones.

Thank you for these adds, sincerely,

Reviewer 2 Report

The manuscript which reports on the fabrication of silica coated cobalt ferrite nanoparticles and their investigation with respect to hyperthermia treatments is well and clearly structured and it is of interest for researchers working in this field.

comments:

Could authors please comment if the influence of the cobalt ferrite particle size is possible.  Did authors perform the VSM measurements at room temperature? -Could be mentioned in the figure caption.  Did authors perform temperature dependent measurements to figure out the blocking temperature of the system? If yes, maye authors could briefly comment. Did authors observe any dipolar magnetic coupling between the particles - due to the thickness of the silica shell it could be possible.
